# Non-invasive evaluation of the equine gastrointestinal mucosal transcriptome

Michelle C. Coleman[ID]<sup></sup>[1‡], Canaan Whitfield-Cargile[ID][1‡*], Noah D. Cohen[ID][1], Jennifer L. Goldsby[2], Laurie Davidson[2], Ana M. Chamoun-Emanuelli[1], Ivan Ivanov[3], Susan Eades[1], Nancy Ing[ID][4], Robert S. Chapkin[2]

**1** Department of Large Animal Clinical Sciences, College of Veterinary Medicine & Biomedical Sciences, Texas A&M University, College Station, Texas, United States of America, **2** Program in Integrative Nutrition & Complex Diseases, College of Agriculture and Life Sciences, Texas A&M University, College Station, Texas, United States America, **3** Department of Veterinary Physiology and Pharmacology, College of Veterinary Medicine & Biomedical Sciences, Texas A&M University, College Station, Texas, United States of America, **4** Department of Animal Science, College of Agriculture and Life Sciences, Texas A&M University, College Station, Texas, United States of America

‡ These authors share first authorship on this work.
* cwhitfield@cvm.tamu.edu

**Data Availability Statement:** The data utilized in this study are available via the NCBI bioproject

## Abstract

Evaluating the health and function of the gastrointestinal tract can be challenging in all species, but is especially difficult in horses due to their size and length of the gastrointestinal (GI) tract. Isolation of mRNA of cells exfoliated from the GI mucosa into feces (*i.e.*, the exfoliome) offers a novel means of non-invasively examining the gene expression profile of the GI mucosa. This approach has been utilized in people with colorectal cancer. Moreover, we have utilized this approach in a murine model of GI inflammation and demonstrated that the exfoliome reflects the tissue transcriptome. The ability of the equine exfoliome to provide non-invasive information regarding the health and function of the GI tract is not known. The objective of this study was to characterize the gene expression profile found in exfoliated intestinal epithelial cells from normal horses and compare the exfoliome data with the tissue mucosal transcriptome. Mucosal samples were collected from standardized locations along the GI tract (*i.e.* ileum, cecum, right dorsal colon, and rectum) from four healthy horses immediately following euthanasia. Voided feces were also collected. RNA isolation, library preparation, and RNA sequencing was performed on fecal and intestinal mucosal samples. Comparison of gene expression profiles from the tissue and exfoliome revealed correlation of gene expression. Moreover, the exfoliome contained reads representing the diverse array of cell types found in the GI mucosa suggesting the equine exfoliome serves as a non-invasive means of examining the global gene expression pattern of the equine GI tract.

## Introduction

Gastrointestinal (GI) disease is of considerable importance to horses and the horse industry, second only to old age as a cause of death [1]. Two decades ago, the cost of colic to the equine

(accession number PRJNA575706) http://www.ncbi.nlm.nih.gov/bioproject.

**Funding:** This project was funded by the Triad for Transformation Program at Texas A&M University, College Station, TX (MC, NI, SE)

**Competing interests:** The authors have declared that no competing interests exist.

industry was $115 million annually, and with continued growth of the equine industry and increasing costs of health care, the staggering financial burden continues to grow [2]. Despite vast research efforts aimed at identifying preventative and treatment strategies for equine GI diseases, they remain a major cause of morbidity and mortality in the horse. The cause of colic in the horse varies considerably, including simple obstructive lesions, strangulating obstructive lesions, and inflammatory conditions. The pathophysiology of these conditions is often poorly understood, resulting in a decreased ability to manage and prevent disease.

An important limitation to understanding the pathogenesis of GI disease and assessing GI health is the lack of non-invasive tools to assess cellular and molecular GI function. Magnetic resonance imaging (MRI) and computed tomography (CT) are frequently utilized in assessing the GI tract in human and small animal medicine, but animal size precludes use of these imaging modalities in horses. Abdominal ultrasonography is widely utilized to examine the equine GI tract and has greatly advanced our ability to accurately diagnose intestinal diseases. Sonographic assessment of the GI tract is limited by the acoustics of the gas-filled intestine [3]. Importantly, irrespective of species, imaging alone does not provide information regarding function of the GI tract at the cellular or molecular level. Currently, intestinal mucosal biopsy is the only available means to provide mechanistic and functional data, but several practical limitations to this approach exist. Endoscopic biopsies can only be obtained from the stomach, duodenum, or rectum. While these biopsies can have diagnostic utility, they do not provide a global view of the GI tract [4]. Samples from other anatomic sites can be acquired via surgical biopsies obtained through traditional open surgical or laparoscopic approaches. Surgical biopsies, however, have several disadvantages including surgical complications, limited ability to biopsy several anatomic locations, and most importantly, the inability to easily obtain longitudinal (sequential) data from individuals regarding intestinal function and health for the purpose of monitoring response to therapy.

Non-invasive coprological assays have been used commonly in people and other animals to diagnose GI disease [5–7]. For example, fecal calprotectin is used to diagnose non-steroidal anti-inflammatory disease (NSAID) enteropathy in people and inflammatory bowel disease (IBD) in dogs [5, 8]. Similar markers of intestinal disease have not been well-studied or validated in horses. Importantly, these are merely markers of inflammation which do not provide mechanistic insight into the cause of the inflammation which would better direct therapeutic interventions. Thus, great clinical and investigative needs exist for the development of non-invasive methods to characterize the health and function of the GI tract to more effectively identify, study, and manage equine intestinal disorders.

A potential strategy to address this limitation is the use of exfoliated intestinal epithelial cells found in feces. Approximately 1/3 of human colonic epithelial cells (up to $10^{10}$ cells in an adult) are exfoliated and shed in the feces daily [9]. A technique to isolate and sequence the mRNA (host transcriptome) from exfoliated intestinal epithelial cells, termed the exfoliome, has been validated in the context of colorectal cancer and neonatal GI development in humans [10–14]. This technique has been utilized in a murine model of NSAID enteropathy, validating its ability to classify animals with GI inflammation [15]. This methodology provides a global view of GI health by assessing the mucosal transcriptome of host cells exfoliated into the GI lumen from the mucosa. To the authors' knowledge, comprehensive evaluation of the transcriptome of the equine GI tract has not been performed. Thus, the objectives of this study were to characterize the gene expression profile found in exfoliated intestinal epithelial cells from normal horses and to compare these data with the tissue mucosal transcriptome from specific locations along the equine GI tract.

## Materials and methods

### Sample population and sample collection

Horses donated to Texas A&M College of Veterinary & Biomedical Sciences for euthanasia for reasons unrelated to GI disease were included in this study. All horses were administered xylazine (AnaSed[®]; 1.1 mg/kg I.V.) and ketamine (Ketaset[®]; 2.2 mg/kg I.V.) prior to euthanasia with potassium chloride (960 mEq I.V.). Feces were collected via rectal palpation immediately following euthanasia, homogenized in RNA Shield[®] (Zymo Research, Irvine, CA), and stored at -80˚C until processed. A ventral midline incision was performed in routine fashion to gain access to the abdomen. The GI tract was exteriorized and within 10 minutes of euthanasia mucosal samples (1 cm x 1 cm) were collected from the ileum, cecum, right dorsal colon, and rectum. RNA isolation, library preparation, and RNA sequencing were performed similarly for fecal and intestinal mucosal samples. This study was approved by the Texas A&M University Institutional Animal Care and Use Committee (IACUC 2016–0301).

### RNA isolation and sequencing from exfoliated cells

PolyA[+] RNA was isolated from fecal samples as previously described [10, 11, 14]. Briefly, RNA was extracted using a commercially available kit (Active Motif, Carslbad, CA), quantified (Nanodrop spectrophotometer; Thermo Fisher Scientific, Waltham, MA), and quality assessed (Bioanalyzer 2100; Agilent Technologies, Santa Clara, CA). Each sample was processed with the NuGen Ovation 3'-DGE kit (San Carlos, CA) to convert RNA into cDNA. Following cDNA fragment repair and purification, Illumina adaptors were ligated onto fragment ends and amplified to create the final library. Libraries were quantified using the NEBNext Library Quant kit for Illumina (NEB, Ipswich, MA) and run on an Agilent DNA High Sensitivity Chip to confirm sizing and the exclusion of adapter dimers.

### RNA isolation from tissue

RNA was extracted from the mucosa of the ileum, cecum, right dorsal colon, and rectum using an E.Z.N.A.[®] Total RNA kit (Omega Bio-tek, Norcross, GA) following the manufacturer's protocol, including on-column DNase treatment. RNA quality was determined using the Nano6000 chip on a Bioanalyzer 2100 (Agilent Technologies). Sequencing libraries were made using 250 ng of RNA and the TruSeq RNA Sample Preparation kit (Illumina) following the manufacturer's protocol.

### Data analysis

The datasets utilized for the current study are available via the NCBI bioproject (accession number PRJNA575706) http://www.ncbi.nlm.nih.gov/bioproject/. Sequencing data were multiplexed and assessed for quality using FastQC. Reads were aligned using Spliced Transcripts Alignment to a reference software with default parameters and referenced against the genome of the horse (EquCab 3.0) [16]. Differentially expressed genes were determined using EdgeR base on the matrix of gene counts [17]. Gene pathway intersections and involvement were analyzed using QIAGEN's Ingenuity Pathway Analysis (IPA, QIAGEN, Redwood City, CA) by uploading gene lists with fold-change and false discovery rate p-values. Statistical analysis was performed using R (v. 3.5.3) statistical software with a level of $P < 0.05$ considered significant.

## Results

### Sample population

Four horses were included in the study including 3 geldings and 1 mare, with a mean age of 12 years (range, 4 to 25 years). There were 3 Quarter Horses and 1 Warmblood included. Reasons for euthanasia included cervical vertebral osteomyelitis (n = 1), equine protozoal myeloencephalitis (n = 1), ocular squamous cell carcinoma (n = 1), and chronic navicular degeneration (n = 1).

### RNA quality from exfoliome versus tissue

RNA quantity and quality were assessed via bioanalyzer for both tissue and feces. The mean RNA integrity number (RIN) from tissue was 8.9 (range, 7.5 to 10) and the mean RIN from fecal samples was 6.2 (range, 5.5 to 6.9). Representative bioanalyzer trace and virtual gel from RNA isolated from equine feces showing expected peaks of 18S and 28S components of the eukaryotic ribosome are shown in S1 Fig. Sequencing quality was assessed by Fastqc as previously described [15, 18]. Representative traces of per base sequence quality demonstrate excellent quality of tissue reads and high quality of exfoliomic reads, albeit more variable and of slightly poorer quality than tissue reads (Fig 1). As has been shown previously, the number of mapped reads was much smaller from the exfoliome than from the tissue [14, 15]. Analysis of the RNA-Seq analysis revealed that there was a greater loss of reads from the exfoliome as compared with the tissue samples (Table 1). Despite loss of reads, there were similar number of genes represented by each of the sample types including the exfoliome (Table 1). The total counts per sample and log(2) counts per gene are shown (S2 Fig).

### Comparison of tissue and exfoliomic data

Genes present in fewer than 2 samples or represented fewer than 10 times across all samples were removed. The intersection of genes represented in the exfoliome and genes represented in each of the tissue samples were calculated (Fig 2). These data indicate that greater than 94% of the genes present in any tissue sample were also represented in the exfoliome. Next, pathways represented by genes present only in the tissue samples and not present in the exfoliome were examined by uploading these genes into Qiagen® IPA software. The pathways represented by these genes are depicted in Table 2.

Interestingly 105 genes were present in the exfoliome, but not identified in any tissue samples (Fig 2). This gene list was analyzed with Qiagen IPA software to identify which pathways were present in the exfoliome but not the tissue samples. The top pathways identified are shown in Table 3.

To further compare the gene expression profiles between the tissue and exfoliome, a principal component analysis (PCA) plot was constructed (Fig 3A). This revealed a visual clustering of exfoliome samples together suggesting a similar gene expression profile of the exfoliome from the four normal horses examined in this study. This PCA also revealed that the tissue samples clustered together and were separated from the exfoliomic samples. To evaluate correlation between the exfoliome and tissue samples, data were normalized with EdgeR *calcnormfactors* using trimmed mean of M-values(TMM). Scatter plots of log(2)-transformed normalized count data between the exfoliome and each tissue source are shown (Fig 3B). There was a strong and significant correlation of all tissues (Spearman's correlation coefficient; $\rho > 0.8$ and $P < 0.0001$) and, although of lesser magnitude, significant correlation between the exfoliome and each tissue source (Spearman's correlation coefficient; $\rho > 0.15$ and $P < 0.0001$) (Table 4).

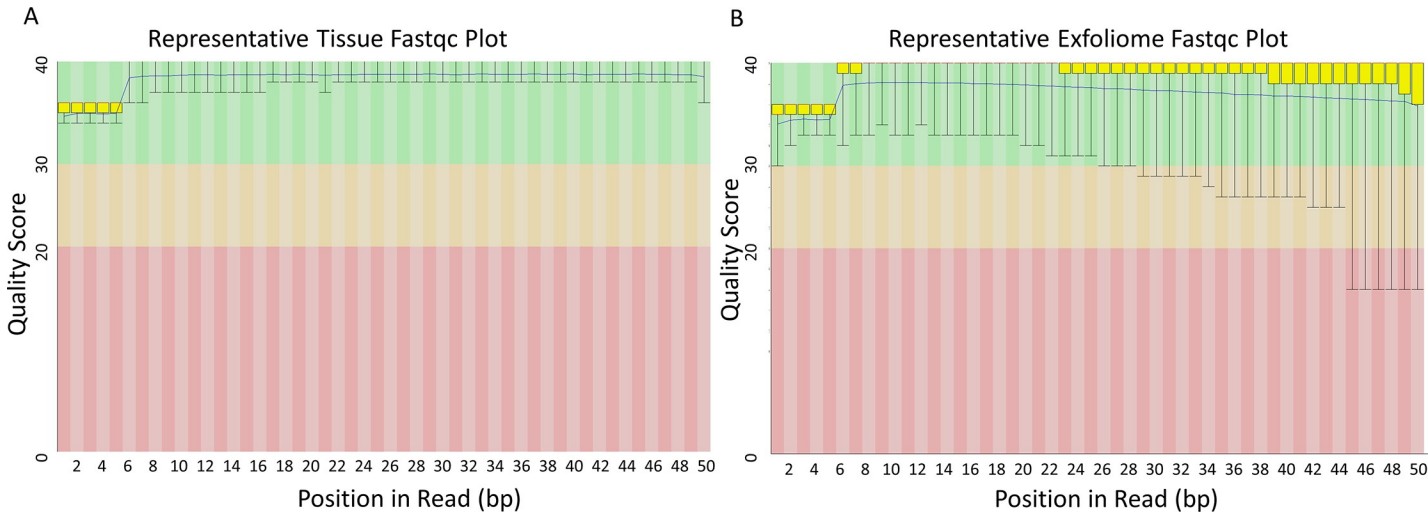

**Fig 1. Sequencing quality of equine exfoliome and mucosal tissue are of excellent quality.** FastQC box plots of quality scores per read position of RNA-Seq data. Y-axis: Phred quality scores 0–40 colored based upon quality of the scores (green = excellent quality (30–40), yellow = acceptable quality (20–30) and red = poor quality 0–20). The yellow area in the box represents the inter-quartile range from the 25th to the 75th percentile. The error bars include the 10th and the 90th percentiles. The red lines inside the box plot are the median value of phred scores for the nucleotide range, and the blue curve represents the mean value. A) Representative fastqc plot from tissue showing quality score per base of fastq sequences. B) Representative fastqc plot from equine exfoliome showing quality score per base of fastq sequences.

## Cell types and anatomic locations represented in the equine exfoliome

It has been previously demonstrated in mice, that the exfoliome gene expression signature arises from multiple anatomic locations and represents a global representation of the GI mucosal transcriptome [15]. In order to determine the source of this signature in horses we extracted the counts of genes previously identified and expressed predominantly in specific anatomic locations (*i.e.*, stomach, small intestine, and colon). Interestingly, we found that the exfoliome contained reads from all major anatomic locations (Fig 4A). As expected, genes representing the colon and small intestine were heavily represented in the transcriptomes arising from those locations with some overlap. Similarly, in addition to anatomic origin, we also assessed the cell types represented in the exfoliome. Clearly, the intestinal epithelium is comprised of many cell types including absorptive cells (enterocytes and colonocytes depending on anatomic location), intestinal stem cells, goblet cells, Paneth cells (SI), among others as well as a host of infiltrating immune cells depending on depth of the sample (*i.e.*, lamina propria) and disease state of the GI tract (*e.g.*, inflammation vs. homeostasis). In order to determine the

**Table 1. Beginning and ending read-counts per sample type at major steps along the analytical RNA-Seq data pipeline.**

|  | Ileum | Cecum | Right Dorsal Colon | Rectum | Exfoliome |
|---|---|---|---|---|---|
| Reads from Sequencing | 26,016,938 (± 5,502,844) | 25,498,238 (± 3,930,189) | 23,449,515 (± 5,862,203) | 24,205,484 (±4,240,166) | 50,336,373 (± 6,269,249) |
| Aligned Reads | 21,693,764 (± 4,464,834) | 20,976,235 (± 2,670,830) | 19,745,684 (± 3,426,658) | 20,857,709 (± 3,760,132) | 15,057,949 (± 2,253,748) |
| No Feature | 6,437,635 (± 1,362,314) | 5,855,954 (± 744,532) | 5,967,331 (± 1,921,276) | 4,998,710 (± 338,597) | 14,758,447 (± 2,215,476) |
| Number of Reads Mapped per Sample | 15,682,057 (± 3,285,220) | 1,512,028 (± 2,481,271) | 12,803,690 (± 3,448,381) | 15,433,069 (± 3,405,894) | 299,502 (± 96,415) |
| Number of Genes Identified | 16,757 (± 135) | 16,793 (± 152) | 16,403 (±472) | 16,098 (± 600) | 14,472 (± 2,391) |

Data represent average per-sample type (± standard deviation).

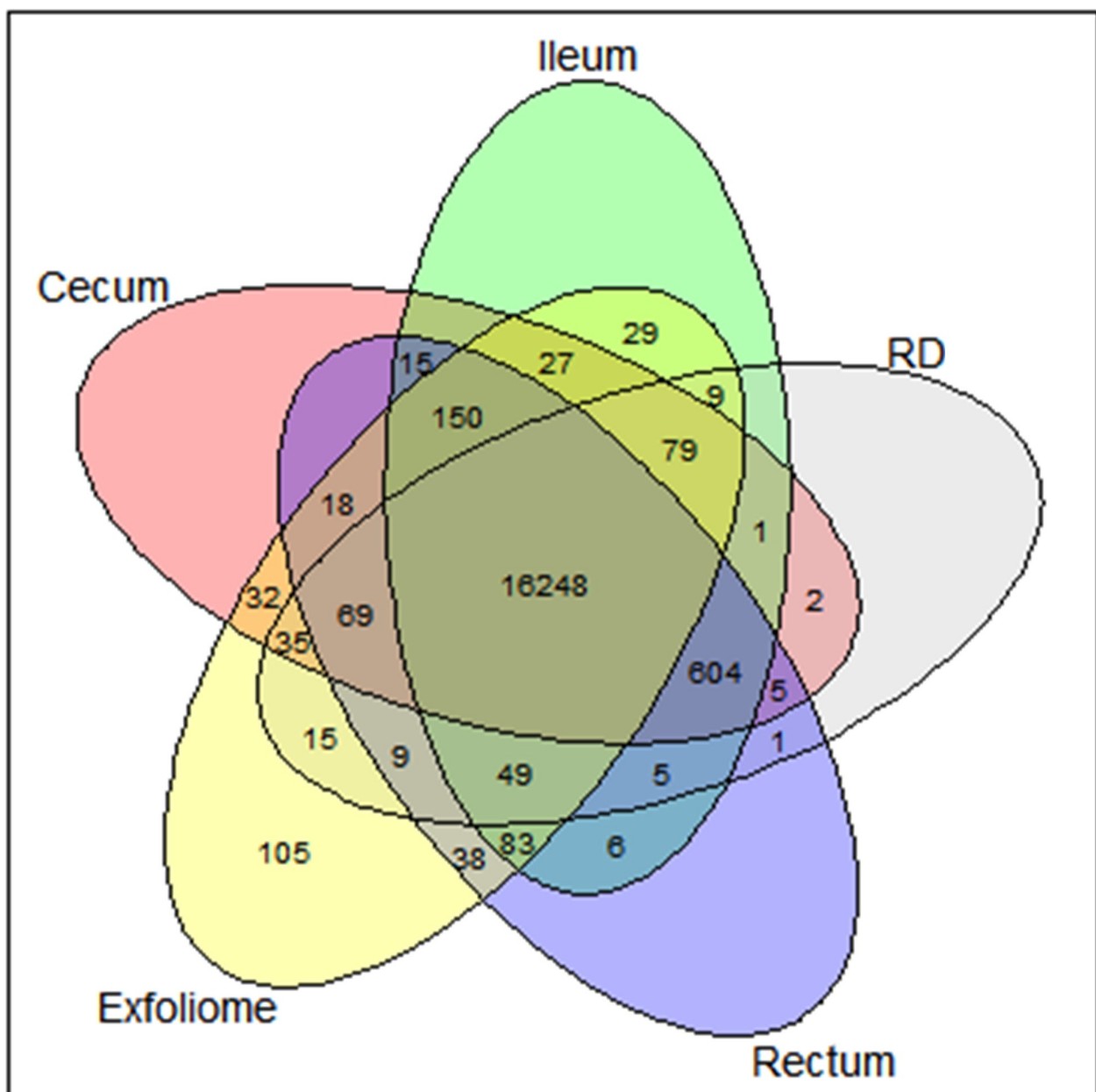

**Fig 2. The equine exfoliome overlaps with the GI mucosal transcriptome from all anatomic sites examined.** Venn diagram depicting the intersection of the exfoliome with the mucosal transcriptome from mucosal biopsies obtained from the ileum, cecum, rectum, and right dorsal colon (RD).

cell types present in these data, we reviewed the literature for marker genes expressed either solely by a specific cell type or at least highly enriched in a specific cell type [19–32]. In particular, we extracted the numbers of reads in each sample for the following cell types: intestinal stem cells, absorptive cells, transit amplifying cells, Paneth cells, tuft cells, goblet cells, macrophages, lymphocytes, neutrophils, and smooth muscle cells. Interestingly, all cell types were present in all datasets as identified by the presence of at least 2 marker genes per cell type (Fig 4B). These data suggest that the equine exfoliome represents gene expression signatures from the diverse array of cell types expected to be found in the intestinal mucosa.

**Table 2. Top canonical pathways enriched by genes found in tissue samples but not found in the equine exfoliome.**

| Canonical Pathway | -log(p-value) of overlap |
|---|---|
| Oxidative Phosphorylation | 7.04 |
| Mitochondrial Dysfunction | 5.70 |
| Granulocyte Adhesion and Diapedesis | 5.43 |
| Regulation of IL-17 by Macrophages | 5.27 |
| Role of Cytokines in Communication Between Immune Cells | 5.01 |
| Atherosclerosis Signaling | 4.81 |
| Agranulocyte Adhesion and Diapedesis | 4.70 |
| LXR/RXR Activation | 4.65 |
| Role of Hypercytokinemia in Pathogenesis | 4.43 |
| Eicosanoid Signaling | 4.05 |
| Communication Between Innate and Adaptive Immunity | 3.35 |
| Role of IL-17A in Inflammation | 3.30 |
| CCR3 Signaling in Eosinophils | 3.24 |

The–log(p-value) of the overlap represents the p-value of the overlap between the inputted gene list and the canonical pathway that is represented.

## Discussion

Despite the presence of degradative host and microbial enzymes in the GI lumen, we were able to, for the first time, extract mRNA from exfoliated intestinal epithelial cells that were voided in equine feces. Further, we have demonstrated that the transcriptome of exfoliated cells in horses 1) represents a similar gene expression profile as the GI tissue transcriptome and 2) represents the multiple anatomic regions of the equine GI tract and all major cell types found in the GI mucosa of healthy horses. Given the limitations of assessing the equine GI tract due to the immense size of horses, this non-invasive approach holds great promise for both research and clinical use.

Exfoliomics has been used in both people and mice to study GI health, response to disease, and effects of therapeutics [10, 11, 14, 15, 33]. There are major physiologic differences between horses and these other species that may have precluded successful use of the approach. For example, the length of the equine GI tract is over 100 feet and GI transit time is up to 48 h in healthy horses [34]. This is vastly different from humans and mice where the GI tract is much shorter and transit time is faster.[35] Degradation of RNA likely occurs when the duration of time between cells exfoliating and voiding of feces is increased. Protection of nucleic acids through proper sample handling is also critical to prevent RNA degradation. While RNA

**Table 3. Canonical pathways enriched by genes found in the equine exfoliome but not found in tissue samples.**

| Canonical Pathway | -log(p-value) of overlap |
|---|---|
| Transcriptional Regulatory Network in Embryonic Stem Cells | 4.91 |
| Gustation Pathway | 2.11 |
| Glutamate Receptor Signaling | 2.00 |
| GABA Receptor Signaling | 1.59 |
| G-Protein Coupled Receptor Signaling | 1.42 |

The–log(p-value) of the overlap represents the p-value of the overlap between the inputted gene list and the canonical pathway that is represented.

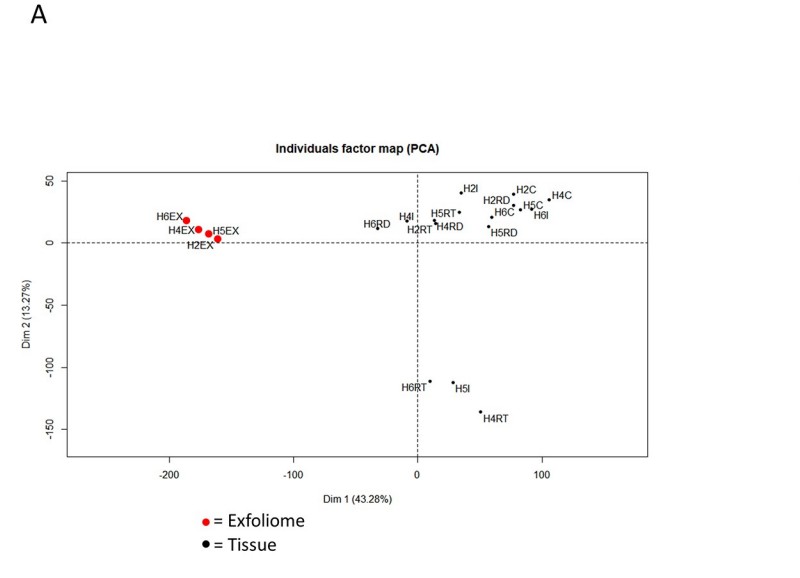

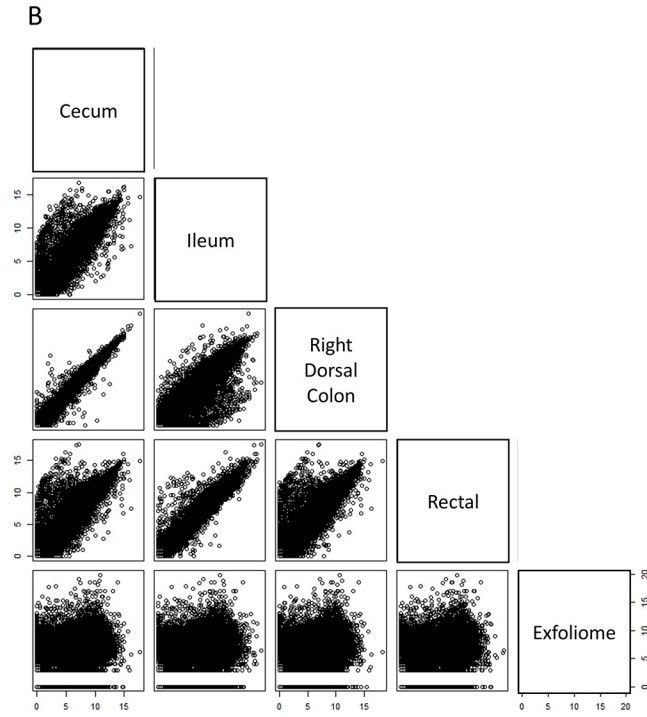

**Fig 3. The equine exfoliome gene expression signature is similar among healthy horses and correlates with the tissue transcriptome.** A) Principal component analysis (PCA) plot showing visual clustering of exfoliome samples suggesting a similar gene expression profile among exfoliome samples from normal horses but distinct from the tissue transcriptome. Samples are identified by sample number (*i.e.*, H2, H4, H5, H6) and sample source (*i.e.* EX = exfoliome, I = ileum, RD = right dorsal colon, C = cecum, RT = rectum). B) Pairwise log(2) scatter plots between each tissue source and the exfoliome showing strong and significant correlation among tissue sites and, to a lesser extent, between tissue sites and the exfoliome.

isolated from exfoliated cells was indeed of lower quality as compared with tissue data, the quality was acceptable and resulted in excellent sequence mapping as compared to human subjects [14].

Our protocol selects for eukaryotic RNA by utilizing oligo dt primers that bind to the polyA tail of eukaryotic transcripts. In people and mice, this approach primarily selects for host mRNA. The microbiota of horses, however, contains vast numbers of eukaryotic organisms. Specifically, as hind-gut fermenters, fermentation in horses is carried out by a host of microorganisms including protozoa in the cecum [36–39]. These protozoa are just one of many eukaryotic organisms found in the equine GI tract. Other types include helminths and fungal organisms, both of which may have been present in high numbers in the horses examined in

**Table 4. Pairwise spearman correlations.**

| Source | Exfoliome | Ileum | Cecum | Right Dorsal Colon | Rectum |
|---|---|---|---|---|---|
| Exfoliome | | 0.166 | 0.186 | 0.155 | 0.157 |
| Ileum | 0.166 | | 0.89 | 0.88 | 0.95 |
| Cecum | 0.186 | 0.89 | | 0.98 | 0.87 |
| Right Dorsal Colon | 0.155 | 0.88 | 0.98 | | 0.86 |
| Rectum | 0.157 | 0.95 | 0.87 | 0.86 | |

Values represent ρ statistic. All correlations were positive and significant (P < 0.0001) although the magnitude of correlation was smaller between the exfoliome and tissues than between the various tissue sites.

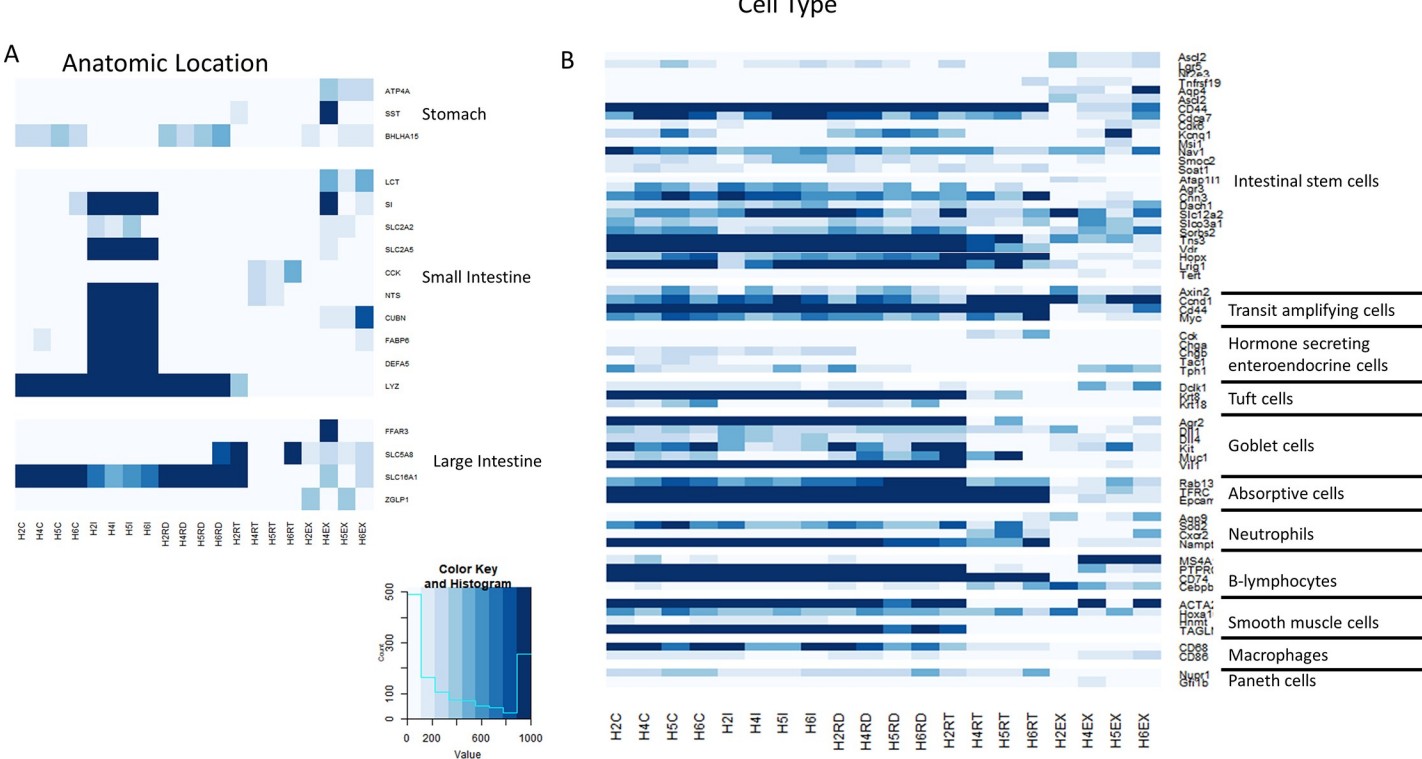

**Fig 4. The equine exfoliome represents all major anatomic locations and cell types found in the equine mucosa.** A) Heatmap depicting frequency of observed counts of genes reported to be predominantly expressed in specific anatomic locations across all samples. B) Heatmap depicting frequency of observed counts of genes reported to be predominantly expressed in specific cell types across all samples. Samples are identified by sample number (*i.e.*, H2, H4, H5, H6) and sample source (*i.e.* EX = exfoliome, I = ileum, RD = right dorsal colon, C = cecum, RT = rectum).

this study. These large numbers of eukaryotic organisms may have explained why only 300,000 to 400,000 reads mapped to the equine genome from a starting number of nearly 50,000,000 reads per sample despite selectively isolating eukaryotic mRNA.

Despite the relatively few number of reads that mapped to the equine genome relative to the tissue samples, this initial evaluation of the equine exfoliome holds promise. Over 94% of the genes present across all tissue samples were present in the equine exfoliome. There were 624 genes present in 2 or more tissue samples and not present in the exfoliome. Interestingly, 10 of the 13 networks enriched by these genes (Table 2) were from inflammatory and immune signaling pathways. Tissue samples were derived from intestinal mucosa obtained via biopsy. The immune cell-rich lamina propria lies just beneath the mucosa and these genes may have been expressed in cells inadvertently obtained from the deeper layers of the intestinal wall. These same cells and genes were unlikely to be expressed in cells exfoliated into the lumen of the GI tract in these healthy horses with no evidence or history of GI disease. Only 105 genes were found in the exfoliome and not in the tissues. Most likely, these genes originated from cells entering the GI lumen and passing into voided stool with intact RNA. Examples of such cells could be derived from the respiratory tract or oral cavity. Only 5 canonical pathways were significantly enriched by these transcripts. Interestingly, the gustation pathway associated with taste was the second most enriched pathway suggesting that indeed these transcripts may have originated from tongue cells that were exfoliated, swallowed, and passed though the GI tract.

Several important limitations of the study should be considered. First, only four horses were included in the study. In addition, there was a great deal of variation in the horses' age (4

to 25 years) and other factors, which may have contributed to some of the variation observed in the exfoliomic signature. Despite this small sample size, we were able to demonstrate that the exfoliated cell transcriptome reflects the tissue-level transcriptome. Another limitation is that these horses had no overt evidence or known history of GI disease, however, it is possible that subclinical or unknown GI disease existed. Importantly, we do not know how concurrent GI disease could affect this technique. Gastrointestinal disease frequently results in inflammation and increased GI transit time. It is unknown if these factors could affect the quality of RNA isolated from exfoliated cells and/or alter biological interpretation as gene expression from these cells could be altered during passage though the GI tract. An important future step will be to examine the equine exfoliome in the context of both health and disease in order to determine if this approach can be used to discriminate healthy from diseased animals and if this approach can be used to gain temporal insight into the pathophysiology of equine GI diseases. There were unexpected gene expression signatures observed at various tissue sites (e.g. Paneth cell markers observed in large intestinal biopsies) and the exact reasons for this unexpected finding are unknown. One possible explanation is that we extrapolated from human and murine data by using genes thought to be predominantly expressed by specific cell types and anatomic locations. However, these same genes may not be specific for locations or cell types in horses. Finally, we compared the exfoliome to the tissue transcriptome at only four anatomic sites. Future work to compare the exfoliome with tissue transcriptome of other sites and especially more proximal sites is important as many diseases specifically affect these locations. Despite these limitations, this is the first work to compare the tissue transcriptome and exfoliome in horses.

## Conclusions

In summary, we have demonstrated that the exfoliated cell global transcriptome closely mirrors the transcriptome of the mucosa of the ileum, right dorsal colon, cecum, and rectum of horses. While the use of exfoliated cells has been validated in other species [12–14], this is the first description of the equine exfoliome and its correlation to the tissue-level transcriptome. Application of this non-invasive technique in early identification or monitoring of GI disease in the horse holds promise, but requires further investigation prior to clinical implementation.

## Supporting information

**S1 Fig. Representative bioanalyzer trace and virtual gel from RNA isolated from equine feces showing expected peaks of 18S and 28S components of the eukaryotic ribosome.** (TIF)

**S2 Fig.** A) Bar plots of total counts per sample from both tissue and feces. B) Boxplots of log2 of gene counts per sample from both tissue and feces. Samples are identified by sample number (*i.e.*, H2, H4, H5, H6) and sample source (*i.e.* EX = exfoliome, I = ileum, RD = right dorsal colon, C = cecum, RT = rectum). (TIF)

## Acknowledgments

The authors acknowledge Texas A&M Institute for Genome Sciences and Society (TIGSS) for providing computational resources and systems administration support for the TIGSS HPC Cluster.

## Author Contributions

**Conceptualization:** Michelle C. Coleman, Canaan Whitfield-Cargile, Noah D. Cohen, Laurie Davidson, Ivan Ivanov, Susan Eades, Robert S. Chapkin.

**Data curation:** Canaan Whitfield-Cargile, Jennifer L. Goldsby, Ana M. Chamoun-Emanuelli.

**Formal analysis:** Canaan Whitfield-Cargile.

**Funding acquisition:** Michelle C. Coleman, Susan Eades, Nancy Ing.

**Investigation:** Michelle C. Coleman, Canaan Whitfield-Cargile, Noah D. Cohen, Jennifer L. Goldsby, Laurie Davidson, Susan Eades, Robert S. Chapkin.

**Methodology:** Canaan Whitfield-Cargile, Noah D. Cohen, Jennifer L. Goldsby, Laurie Davidson, Ivan Ivanov, Susan Eades, Nancy Ing, Robert S. Chapkin.

**Project administration:** Canaan Whitfield-Cargile.

**Resources:** Canaan Whitfield-Cargile, Jennifer L. Goldsby, Laurie Davidson, Susan Eades, Robert S. Chapkin.

**Software:** Canaan Whitfield-Cargile, Jennifer L. Goldsby, Robert S. Chapkin.

**Supervision:** Robert S. Chapkin.

**Writing – original draft:** Michelle C. Coleman, Canaan Whitfield-Cargile.

**Writing – review & editing:** Michelle C. Coleman, Noah D. Cohen, Jennifer L. Goldsby, Laurie Davidson, Ana M. Chamoun-Emanuelli, Ivan Ivanov, Susan Eades, Nancy Ing, Robert S. Chapkin.

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
