## [Decision Letter · Decision Letter 0]

4 Dec 2019

PONE-D-19-27855

Non-invasive evaluation of the equine gastrointestinal mucosal transcriptome

PLOS ONE

Dear Dr. Whitfield-Cargile,

Thank you for submitting your manuscript to PLOS ONE. After careful consideration, we feel that it has merit but does not fully meet PLOS ONE’s publication criteria as it currently stands. Therefore, we invite you to submit a revised version of the manuscript that addresses the points raised during the review process.

ACADEMIC EDITOR: 

The authors have provided some valuable RNA seq data obtained using exfoliate and GI mucosal samples from healthy horses. They have compared these RNA seq and report a relatively strong correlation between them. The authors try to establish a non-invaisive methodology to obtain global gene expression pattern unique to GI tract. As suggested by reviewers 2 and 3, the correlation does not appear to be strong. This is a well-written manuscript and I would suggest the authors to respond to reviewers comments to increase the impact of the manuscript.

We would appreciate receiving your revised manuscript by Jan 18 2020 11:59PM. To enhance the reproducibility of your results, we recommend that if applicable you deposit your laboratory protocols in protocols.io, where a protocol can be assigned its own identifier (DOI) such that it can be cited independently in the future. For instructions see: http://journals.plos.org/plosone/s/submission-guidelines#loc-laboratory-protocols

We look forward to receiving your revised manuscript.

Kind regards,

Sripathi M Sureban, Ph.D.

Academic Editor

PLOS ONE

Journal Requirements:

2. Please amend either the abstract on the online submission form (via Edit Submission) or the abstract in the manuscript so that they are identical.

3.  We note that you are reporting an analysis of a microarray, next-generation sequencing, or deep sequencing data set. PLOS requires that authors comply with field-specific standards for preparation, recording, and deposition of data in repositories appropriate to their field. Please upload these data to a stable, public repository (such as ArrayExpress, Gene Expression Omnibus (GEO), DNA Data Bank of Japan (DDBJ), NCBI GenBank, NCBI Sequence Read Archive, or EMBL Nucleotide Sequence Database (ENA)). In your revised cover letter, please provide the relevant accession numbers that may be used to access these data. For a full list of recommended repositories, see http://journals.plos.org/plosone/s/data-availability#loc-omics or http://journals.plos.org/plosone/s/data-availability#loc-sequencing.

Reviewers' comments:

Reviewer's Responses to Questions

**Comments to the Author**

1. Is the manuscript technically sound, and do the data support the conclusions?

Reviewer #1: Yes

Reviewer #2: Yes

Reviewer #3: Partly

2. Has the statistical analysis been performed appropriately and rigorously? 

Reviewer #1: Yes

Reviewer #2: I Don't Know

Reviewer #3: Yes

3. Have the authors made all data underlying the findings in their manuscript fully available?

Reviewer #1: Yes

Reviewer #2: Yes

Reviewer #3: Yes

4. Is the manuscript presented in an intelligible fashion and written in standard English?

Reviewer #1: Yes

Reviewer #2: Yes

Reviewer #3: Yes

5. Review Comments to the Author

Reviewer #1: This study provides a very logical and detailed evaluation of how the exfoliome relates to the transcriptome. I have no substantial edits or comments to revise the paper and I am in support of its publication.

1. Line 38: Typo “healthy” horses.

2. Line 109: Typo “Irvine,” CA.

Reviewer #2: Overall this reviewer believes that this is a well written manuscript on a timely and interesting topic. However, there appears to be confusion as to the most recently accepted terminology for epithelial cell identity, particularly within the crypt base. A point by point review can be found below.

Abstract

Line 37-39: Please revise sentence. As written is appears that feces was directly collected from all sites in addition to mucosal samples. Line 38 misspelling ‘four healthy’ not ‘health’ horses.

Introduction

Line 63-64: Vague statement, I would recommend just stating what the main limitation(s) of ultrasound in the horse are. This is not an equine journal and it cannot be assumed that the average reader would know.

Lines 67-77: I found these lines implicit not explicitly written. Furthermore, if a sentence starts with ‘first’ then a following sentence should start with ‘second’. It is not obvious why endoscopic biopsies are inferior to “other” types of biopsies. This is inferred in the following sentences and this reviewer agrees with this statement however I am not sure the average reader would know the difference. I believe that this entire section should be re-written to make the description of limitations of each biopsy technique more clear and more concise.

Line 82-84: It is not clear what the author means by ‘mechanistic’? Identifying intestinal inflammation can be of diagnostic value and important for direct clinical interventions particularly because current clinical therapies for intestinal disease are limited. However, I would agree that diagnosing dysfunction more specifically may help direct future therapies.

Line 94-95: “global” is used twice in this sentence, please consider using another term

Line 96: The word transcriptome should be changed to exfoliome or mucosal transcriptome? Or both?

Line 98: The term exfoliome should be defined within the manuscript earlier and then used throughout the introduction.

Results

Table 1: I don’t believe the abbreviation for right dorsal colon (RDC) is stated within the text prior to use within the table. Please be consistent throughout using either RD or RDC for right dorsal colon. Rectal should be changed to Rectum.

Table 4: Exfoliome doesn’t appear to be very well correlated with mucosa from any section of the intestine although I appreciate that the values are positive? This is, however, stated very strongly throughout the manuscript.

Line 245-247: Stem cells, goblet cells and Paneth cells are all examples of intestinal epithelial cells, along with enterocytes, enteroendocrine and tuft cells. This sentence is written as if the author is not aware of all of the cell types that make up the intestinal epithelium? Please clarify.

Line 246: How is the author defining ‘crypt cells’? Do they mean transit amplifying cells? The crypt arguably consists of stem, Paneth/Paneth-like cells, goblet cells, and transit amplifying cells (small intestine versus colon).

Line 251: Should reference Gonzalez et al AJVR, particularly because this manuscript specifically identifies biomarkers in equine GI derived tissues.

Line 252: See above. Not sure what the author is meaning to represent with IECs? Furthermore, intestinal epithelial stem cells are commonly called crypt based columnar stem cells. Again, not sure what cell types the authors are trying to distinguish. Please clarify, as written there appears to be confusion as to all of the cell types and the nomenclature used for the types of intestinal epithelial cell types.

Line 265: Be consistent through manuscript. RD or RDC to represent the right dorsal colon.

Discussion:

There is a general lack of discussion on the pathways listed in Table 2. There doesn’t appear to be a point in listing the specific Canonical pathways. Although the reviewer finds these pathways to be interesting there is no discussion as to the significance of these findings.

The authors have failed to fully describe other limitations of this technique. Many times in disease GI motility is decreased or absent. How would this could further delay/impact the quality of RNA collected? Does the author propose a way to distinguish different segments of the bowel? It is unfortunate that the authors only chose to sample from the ileum as opposed to other segments of the small intestine such as the duodenum and jejunum where many inflammatory forms of enteritis arise. Similar comment with regard to the different segments of the colon.

Figure 1 Legend: Please better define what the yellow squares with associated error bars represent?

Supplemental Figure 2 Legend: Please include what the labeling on the x axis represents. Every figure and figure legend should stand alone and be defined.

Figure 4: Grouping of cells types is confusing on right side of figure. Normally the acronym CBC is used for crypt based columnar stem cells. If the author is generally grouping all crypt based columnar cells (not just stem cells) those would include intestinal stem cells, Paneth cells and transit amplifying cells (if small intestine). I would argue that hallmark biomarkers for crypt based columnar stem cells include Lgr5, Ascl2, Olfm4 to name a few (but these are the major ones). There is overlap between biomarker expression throughout the crypt and between active and reserve (quiescent) stem cells. Please review the literature for the most recent accepted nomenclature for the cells comprising the intestinal epithelial crypt.

With regard to the data represented in Fig 4B, there appears to be low expression of Paneth cell markers in the small intestinal samples and higher expression in the right dorsal colon and even the rectum in one horse? This is strange considering Paneth cells aren’t thought to exist in the colon? A comment on this is recommended.

Reviewer #3: The study is the first to compare the tissue and exfoliome transcriptomes in horses. The paper flows quite nicely, using appropriate lab techniques and statistical methods for data analysis. Conclusions are supported by their data.

Major comments:

1. Small sample size (n=4) and heterogeneous study subjects. Although all called "healthy" horses, they differ in terms of age and reasons for euthanasia. With such a small sample size, it is difficult to generalize the study findings to healthy horses at large.

2. The Spearman correlation coefficients between exfoliome and each tissue source range from 0.155 to 0.186. These are only considered week associations, albeit statistical significance. Also, the corresponding scatter plots did not show a clear positive correlation. Therefore, potential clinical usage of exfoliome is not very convincing. It would have been a much stronger study if the authors included both healthy and GI diseased horses and demonstrated difference between the groups using exfoliome.

Minor comments:

1. All the figures are of low resolution and very hard to see.

2. line 38: change 'health' to 'healthy'

3. line 214: define 'TMM'

6. PLOS authors have the option to publish the peer review history of their article (what does this mean?). If published, this will include your full peer review and any attached files.

Reviewer #1: No

Reviewer #2: No

Reviewer #3: No

---

## [Author Response · Author response to Decision Letter 0]

8 Jan 2020

January 8, 2020

Editorial Board of PlosOne

Subject: Manuscript entitled, “Non-invasive evaluation of the equine gastrointestinal mucosal transcriptome” (PONE-D-19-27855)

Dear Members of the Editorial and Review Board:

Thank you sincerely for your careful, thorough and thoughtful review of the manuscript. We have carefully considered and responded to each of the recommendations. It is our opinion that through this review process the clarity and quality of the manuscript has substantially improved. Each comment posed by the reviewers had been addressed below with corresponding changes and improvements to the manuscript. 

Reviewer #1: 

This study provides a very logical and detailed evaluation of how the exfoliome relates to the transcriptome. I have no substantial edits or comments to revise the paper and I am in support of its publication.

Line 38: Typo “healthy” horses.

Authors’ Response: Corrected. Thank you

Line 109: Typo “Irvine,” CA.

Authors’ Response: Corrected. Thank you

Reviewer #2: 

Overall this reviewer believes that this is a well written manuscript on a timely and interesting topic. However, there appears to be confusion as to the most recently accepted terminology for epithelial cell identity, particularly within the crypt base. A point by point review can be found below.

Abstract

Line 37-39: Please revise sentence. As written is appears that feces was directly collected from all sites in addition to mucosal samples. Line 38 misspelling ‘four healthy’ not ‘health’ horses.

Authors’ Response: The structure and grammar of the sentence was improved. 

Introduction

Line 63-64: Vague statement, I would recommend just stating what the main limitation(s) of ultrasound in the horse are. This is not an equine journal and it cannot be assumed that the average reader would know.

Authors’ Response: This sentence was re-worded as requested

Lines 67-77: I found these lines implicit not explicitly written. Furthermore, if a sentence starts with ‘first’ then a following sentence should start with ‘second’. It is not obvious why endoscopic biopsies are inferior to “other” types of biopsies. This is inferred in the following sentences and this reviewer agrees with this statement however I am not sure the average reader would know the difference. I believe that this entire section should be re-written to make the description of limitations of each biopsy technique more clear and more concise.

Authors’ Response: This section was clarified.

Line 82-84: It is not clear what the author means by ‘mechanistic’? Identifying intestinal inflammation can be of diagnostic value and important for direct clinical interventions particularly because current clinical therapies for intestinal disease are limited. However, I would agree that diagnosing dysfunction more specifically may help direct future therapies.

Authors’ Response: “Mechanistic” in this sentence refers to the mechanism by which inflammation occurred. This has been reworded for clarification.

Line 94-95: “global” is used twice in this sentence, please consider using another term

Authors’ Response: Corrected

Line 96: The word transcriptome should be changed to exfoliome or mucosal transcriptome? Or both?

Authors’ Response: Corrected

Line 98: The term exfoliome should be defined within the manuscript earlier and then used throughout the introduction.

Authors’ Response: We have defined exfoliome at the first introduction of the concept of utilizing the transcriptome of exfoliated cells.

Results

Table 1: I don’t believe the abbreviation for right dorsal colon (RDC) is stated within the text prior to use within the table. Please be consistent throughout using either RD or RDC for right dorsal colon. Rectal should be changed to Rectum.

Authors’ Response: Corrected

Table 4: Exfoliome doesn’t appear to be very well correlated with mucosa from any section of the intestine although I appreciate that the values are positive? This is, however, stated very strongly throughout the manuscript.

Authors’ Response: This has been modified as requested. In all locations where the degree of correlation between tissue and exfoliome is stated we make it clear that this correlation was positive and significant but to a lesser extent than the correlation between tissue sites. This includes Figure 3 and Table 4 legends as well as the body of the manuscript.

Line 245-247: Stem cells, goblet cells and Paneth cells are all examples of intestinal epithelial cells, along with enterocytes, enteroendocrine and tuft cells. This sentence is written as if the author is not aware of all of the cell types that make up the intestinal epithelium? Please clarify.

Authors’ Response: Corrected

Line 246: How is the author defining ‘crypt cells’? Do they mean transit amplifying cells? The crypt arguably consists of stem, Paneth/Paneth-like cells, goblet cells, and transit amplifying cells (small intestine versus colon).

Authors’ Response: We apologize for this poorly worded sentence and mistakenly included ‘crypt cells’. This sentence has been revised to reflect the fact that the intestinal mucosa houses a diverse array of cell types. 

Line 251: Should reference Gonzalez et al AJVR, particularly because this manuscript specifically identifies biomarkers in equine GI derived tissues.

Authors’ Response: The reference was added. Thank you for this suggestion. 

Line 252: See above. Not sure what the author is meaning to represent with IECs? Furthermore, intestinal epithelial stem cells are commonly called crypt based columnar stem cells. Again, not sure what cell types the authors are trying to distinguish. Please clarify, as written there appears to be confusion as to all of the cell types and the nomenclature used for the types of intestinal epithelial cell types.

Authors’ Response: We were attempting to distinguish expression patterns of genes in LGR5+ cells along the crypt as has been described previously (“The Lgr5 intestinal stem cell signature: robustexpression of proposed quiescent ‘+4’ cell markers” Munoz et. al 2012) and to distinguish CBC cells from the proposed quiescent +4 cell population that is purported to replenish CBC cells in response to injury. We agree, however, that we did a poor job of clarifying these cell types. Moreover, we recognize that this is a controversial topic and beyond the scope of this manuscript. Therefore, we have revised the text and figures to simply represent intestinal stem cell markers without attempting to identify subtypes which is controversial in any species and unstudied in the horse. 

Line 265: Be consistent through manuscript. RD or RDC to represent the right dorsal colon.

Authors’ Response: We have removed RDC throughout the manuscript and figures and right dorsal colon is now consistently represented as RD.

Discussion:

There is a general lack of discussion on the pathways listed in Table 2. There doesn’t appear to be a point in listing the specific Canonical pathways. Although the reviewer finds these pathways to be interesting there is no discussion as to the significance of these findings.

Authors’ Response: The pathways identified in table 2 are discussed in lines 312-319 of the discussion. We have edited this section to make it more clear that this paragraph refers to table 2. 

The authors have failed to fully describe other limitations of this technique. Many times in disease GI motility is decreased or absent. How would this could further delay/impact the quality of RNA collected? Does the author propose a way to distinguish different segments of the bowel? It is unfortunate that the authors only chose to sample from the ileum as opposed to other segments of the small intestine such as the duodenum and jejunum where many inflammatory forms of enteritis arise. Similar comment with regard to the different segments of the colon.

Authors’ Response: We agree that there are many limitations to this technique. We agree that GI disease may affect the quality and biological interpretation of exfoliomics and this has been added to the discussion as requested. This approach provides a global view of mucosal genes expression as stated and, currently, distinguishing different segments of the bowel is not possible. We agree that sampling and sequencing more anatomic locations would improve this manuscript but that was beyond the scope of the current work. Future work will aim to address these limitations and this has been added to the discussion. 

Figure 1 Legend: Please better define what the yellow squares with associated error bars represent?

Authors’ Response: The fastqc plots have been defined as requested.

Supplemental Figure 2 Legend: Please include what the labeling on the x axis represents. Every figure and figure legend should stand alone and be defined.

Authors’ Response: The legend has been corrected. 

Figure 4: Grouping of cells types is confusing on right side of figure. Normally the acronym CBC is used for crypt based columnar stem cells. If the author is generally grouping all crypt based columnar cells (not just stem cells) those would include intestinal stem cells, Paneth cells and transit amplifying cells (if small intestine). I would argue that hallmark biomarkers for crypt based columnar stem cells include Lgr5, Ascl2, Olfm4 to name a few (but these are the major ones). There is overlap between biomarker expression throughout the crypt and between active and reserve (quiescent) stem cells. Please review the literature for the most recent accepted nomenclature for the cells comprising the intestinal epithelial crypt.

Authors’ Response: As stated above, we agree that this area is controversial and our goal of attempting to distinguish expression patterns of genes in LGR5+ cells along the crypt and to distinguish CBC cells from the proposed quiescent +4 cell population is beyond the scope of this manuscript. Therefore, we have revised the text and figures to simply represent intestinal stem cell markers without attempting to identify subtypes which is controversial in any species and unstudied in the horse. 

With regard to the data represented in Fig 4B, there appears to be low expression of Paneth cell markers in the small intestinal samples and higher expression in the right dorsal colon and even the rectum in one horse? This is strange considering Paneth cells aren’t thought to exist in the colon? A comment on this is recommended.

 Authors’ Response: A comment on this has been added in the discussion as requested. While the exact reasons for the unexpected expression of purported marker genes (e.g. Paneth cell markers in the large intestine) is unknown, one possible reason is that many of these markers are not only expressed by the cell type to which they are reported to be predominantly expressed by. For example, lysozyme is widely accepted to be a marker for Paneth cells yet this enzyme is also expressed in neutrophils and macrophages. Similarly, the specificity for many of these markers is not known in the horse and perhaps other cell types may express these markers thereby explaining differences in observed expression along the anatomic sites examined compared with expected expression in these locations. 

Reviewer #3: 

The study is the first to compare the tissue and exfoliome transcriptomes in horses. The paper flows quite nicely, using appropriate lab techniques and statistical methods for data analysis. Conclusions are supported by their data.

Major comments:

Small sample size (n=4) and heterogeneous study subjects. Although all called "healthy" horses, they differ in terms of age and reasons for euthanasia. With such a small sample size, it is difficult to generalize the study findings to healthy horses at large. 

Authors’ Response: We appreciate your comments and suggestions, and agree that our sample size is small and study population is diverse. However, this is a proof-of-principle study to demonstrate the possible utility of a novel technique. Results of this study should not be extroplolated to other populations (i.e. diseased horses). Further studies with better defined populations is warranted. The discussion section has been improved to reflect this concern.

The Spearman correlation coefficients between exfoliome and each tissue source range from 0.155 to 0.186. These are only considered week associations, albeit statistical significance. Also, the corresponding scatter plots did not show a clear positive correlation. Therefore, potential clinical usage of exfoliome is not very convincing. It would have been a much stronger study if the authors included both healthy and GI diseased horses and demonstrated difference between the groups using exfoliome.

Authors’ Response: As stated above, the next step will be to explore the utility of this technique in horses with well-defined GI disease. Clarification of the correlation of exfoliome and each tissue source has been added to the text. 

Minor comments

All the figures are of low resolution and very hard to see.

Authors’ Response: We apologize for this and agree that figures are difficult to see on the proof. It is our understanding that this is due to low quality images being included in the proof as a default of the manuscript submission software. The actual figures are > 300 dpi and pass the manuscript submissions software quality check. 

Line 38: change 'health' to 'healthy'

Authors’ Response: Corrected

Line 214: define 'TMM'

Authors’ Response: Corrected

 We again thank you for the time and effort dedicated to this review process. We hope you agree that the clarity and quality of the manuscript has substantially improved.

---

## [Decision Letter · Decision Letter 1]

5 Feb 2020

PONE-D-19-27855R1

Non-invasive evaluation of the equine gastrointestinal mucosal transcriptome

PLOS ONE

Dear Dr. Whitfield-Cargile,

Thank you for submitting your manuscript to PLOS ONE. After careful consideration, we feel that it has merit but does not fully meet PLOS ONE’s publication criteria as it currently stands. Therefore, we invite you to submit a revised version of the manuscript that addresses the points raised during the review process.

ACADEMIC EDITOR: This is a very good study and the revised manuscript is significantly better than the previous versions. Please address the minor concerns raised by the review #2. 

We would appreciate receiving your revised manuscript by Mar 21 2020 11:59PM. To enhance the reproducibility of your results, we recommend that if applicable you deposit your laboratory protocols in protocols.io, where a protocol can be assigned its own identifier (DOI) such that it can be cited independently in the future. For instructions see: http://journals.plos.org/plosone/s/submission-guidelines#loc-laboratory-protocols

We look forward to receiving your revised manuscript.

Kind regards,

Sripathi M Sureban, Ph.D.

Academic Editor

PLOS ONE

Reviewers' comments:

Reviewer's Responses to Questions

**Comments to the Author**

1. If the authors have adequately addressed your comments raised in a previous round of review and you feel that this manuscript is now acceptable for publication, you may indicate that here to bypass the “Comments to the Author” section, enter your conflict of interest statement in the “Confidential to Editor” section, and submit your "Accept" recommendation.

Reviewer #1: All comments have been addressed

Reviewer #2: All comments have been addressed

Reviewer #3: All comments have been addressed

2. Is the manuscript technically sound, and do the data support the conclusions?

Reviewer #1: Yes

Reviewer #2: Yes

Reviewer #3: Yes

3. Has the statistical analysis been performed appropriately and rigorously? 

Reviewer #1: Yes

Reviewer #2: Yes

Reviewer #3: Yes

4. Have the authors made all data underlying the findings in their manuscript fully available?

Reviewer #1: Yes

Reviewer #2: Yes

Reviewer #3: Yes

5. Is the manuscript presented in an intelligible fashion and written in standard English?

Reviewer #1: Yes

Reviewer #2: Yes

Reviewer #3: Yes

6. Review Comments to the Author

Reviewer #1: (No Response)

Reviewer #2: Overall, this is a significantly improved manuscript. The authors have addressed almost all of this reviewer's concerns.

A few points that need to be addressed.

Abstract:

Line 41: There was a coorelation, questionable if this could be stated as 'strong', please revise.

Intro: Overall, a much nicer read!

Results:

Line 223: This statement is what is mis-represented in the abstract. As written, “There was a strong and significant coorelation of all tissues…an, although of lesser magnitude, significant correlation between the exfoliome and each tissue source”.

Reviewer #3: (No Response)

7. PLOS authors have the option to publish the peer review history of their article (what does this mean?). If published, this will include your full peer review and any attached files.

Reviewer #1: No

Reviewer #2: No

Reviewer #3: No

---

## [Author Response · Author response to Decision Letter 1]

12 Feb 2020

February 12th, 2020

Editorial Board of PlosOne

Subject: Manuscript entitled, “Non-invasive evaluation of the equine gastrointestinal mucosal transcriptome” (PONE-D-19-27855R1)

Dear Members of the Editorial and Review Board:

Thank you for your careful, thorough and thoughtful review of the manuscript. We have carefully considered and responded to each of the recommendations. Each comment posed by the reviewers had been addressed below with corresponding changes and improvements to the manuscript. 

Reviewer #1: 

No response

Reviewer #2:

Overall, this is a significantly improved manuscript. The authors have addressed almost all of this reviewer's concerns.

A few points that need to be addressed.

Abstract:

Line 41: There was a coorelation, questionable if this could be stated as 'strong', please revise.

Authors’ Response: This was revised as requested. We apologize for missing this over statement in the initial revision. 

Results:

Line 223: This statement is what is mis-represented in the abstract. As written, “There was a strong and significant coorelation of all tissues…an, although of lesser magnitude, significant correlation between the exfoliome and each tissue source”.

Authors’ Response: This was revised in the abstract as requested. 

Reviewer #3: 

No response

We again thank you for the time and effort dedicated to this review process. We hope you agree that the clarity and quality of the manuscript has substantially improved. For any questions or concerns regarding this submission, please contact me by telephone at my office (979-845-3541) or by email (cwhitfield@cvm.tamu.edu). Thank you for your consideration.

---

## [Editor Report · Decision Letter 2]

14 Feb 2020

Non-invasive evaluation of the equine gastrointestinal mucosal transcriptome

PONE-D-19-27855R2

Dear Dr. Whitfield-Cargile,

We are pleased to inform you that your manuscript has been judged scientifically suitable for publication and will be formally accepted for publication once it complies with all outstanding technical requirements.

With kind regards,

Sripathi M Sureban, Ph.D.

Academic Editor

PLOS ONE
---

## [Editor Report · Acceptance letter]

3 Mar 2020

PONE-D-19-27855R2 

Non-invasive evaluation of the equine gastrointestinal mucosal transcriptome 

Dear Dr. Whitfield-Cargile:

I am pleased to inform you that your manuscript has been deemed suitable for publication in PLOS ONE. Congratulations! Your manuscript is now with our production department. 

With kind regards,

on behalf of

Dr. Sripathi M Sureban 

Academic Editor

PLOS ONE